# The Virtual Citizen Science Expo Hall: A Case Study of a Design-Based Project for Sustainability Education

**Tutaleni I. Asino** [1,*], **Nicole M. Colston** [2], **Ayodeji Ibukun** [1] and **Clement Abai** [1]

1    Emerging Technologies and Creativity Research Lab, Learning, Design and Technology Program, College of Education and Human Science, Oklahoma State University, Stillwater, OK 74078, USA; ayo.ibukun@okstate.edu (A.I.); clement.abai@okstate.edu (C.A.)

2    Natural Resource Ecology & Management, Division of Agricultural Sciences and Natural Resources, Oklahoma State University, Stillwater, OK 74078, USA; nicole.colston@okstate.edu

\*    Correspondence: tutaleni.asino@okstate.edu

**Abstract:** A design-based project grounded in learning technology theories and systematically implemented can impact environmental education in many positive ways. This paper explores the systematic application of best practices from design-based projects that were used to combine and implement a drought education program. Embracing diffusion of innovation as its framework, augmented and virtual reality applications were used to design a virtual meeting space called the Virtual Citizen Science Expo. The results and findings show that users found Mozilla Hubs engaging as it gave them new ideas on the creative and inspirational use of virtual reality technology as an interactive and collaborative learning space. The discussions demonstrate that our VCSE can be used to promote and engage learners in science related to environmental monitoring.

**Keywords:** sustainability; learning/educational technologies; informal science learning; virtual reality; augmented reality; citizen science; drought monitoring

## 1. Introduction

In 2015, the United Nations (UN) adopted the Sustainable Development Goals (SDGs) Agenda 2030, in which sustainable educational technology tools and practices are described as part of the foundations necessary to build upon innovations and infrastructures of learning [1,2]. These tools and practices can be described as either digital technological innovations or technological integrations tied to change processes and approaches [3]. Three years later, in 2018, UNESCO identified trends in education for sustainable development and reaffirmed the use of Information and communication technologies (ICTs) to scale up sustainability education. One goal of education for sustainable development (ESD) is for citizens to be able to use technology broadly and interactively [4]. Immersive technologies, such as augmented reality (AR), virtual reality, and artificial intelligence (AI), offer new opportunities for education and engagement. Hence, there is a need to understand how these emerging learning technologies support ESD, which is the contribution of this article.

This paper shares the systematic application of best practices from design-based projects used to implement a Virtual Citizen Science program focusing on the sustainability issue of drought. The following section shares the literature review that looks at the research connecting citizen science and learning technologies for sustainability and specifically supports VR as a learning tool. Subsequently, we talk about the theoretical framework and background where we showed how diffusion of the innovation theory informed the background of our study. Next, we discuss the method, showing the procedures for designing the VCSE. The results and findings illustrate how we have used learning technologies for ESD. The discussion section mentioned important details of interactions observed and recorded in the VCSE. Finally, in the conclusion section, we tie all the pieces of our study together and provide our recommendations for future studies.

## 2. Literature Review

Virtual reality is a method by which real and virtual environments are interlaced to produce digital content [5]. VR can be useful as a venue for providing informal science learning experiences. Informal science learning refers to lifelong learning that is unstructured, participant-driven, and collaborative. Emerging technologies, such as VR, present a great potential to recruit, develop, and sustain diverse participants' interest and learning in science, math, and engineering (STEM) [6,7] in learning environments, such as museums, science festivals, afterschool programs, and libraries. These learning spaces present authentic and direct pathways for science education, science communication, and public outreach [8]. However, before delving further into the VR space, a definition of learning technologies is necessary to situate how we approach this topic.

### 2.1. Defining Learning Technologies

Technologies have been influencing and supporting learning since time immemorial. For example, the invention of writing to the printing press, significantly impacted learning and sustaining of knowledges globally. Technologies used for learning continue to evolve, spanning formal and informal learning practices. However, as technologies continue to develop and change, so does our definition and conceptualization of the concept. Before delving into how learning technologies support learning about sustainability, it is necessary to define the term in the context of this paper. Definitions of learning technologies range, from focusing on the technology to emphasizing support of the process of learning. Table 1 below presents a sample of definitions that are widely in use.

**Table 1.** A Sample of Learning/Educational Technology Definitions.

| The Association for Educational Communications and Technology (AECT) | Association for Learning Technology | Encyclopedia of the Sciences of Learning |
|---|---|---|
| "Educational technology is the study and ethical application of theory, research, and best practices to advance knowledge as well as mediate and improve learning and performance through the strategic design, management and implementation of learning and instructional processes and resources" [9] | We define learning technology as "the broad range of communication, information and related technologies that are used to support learning, teaching and assessment" [10]. | Learning technology refers to a field of study and practices of technology for learning and technology of learning. "Technology for learning pertains to the use of technology during (the support of) learning processes...." "Technology of learning" relates to the question on how scientific findings with respect to (supporting) learning can actually be used to support learning processes" [11]. |

This study explored augmented reality (AR) and virtual reality (VR) as learning technologies to support ESD. Specifically, in this paper, we report on the using of a virtual reality space to support learning about drought impacts and the importance of local environmental monitoring for community resilience.

### 2.2. Agricultural Drought in the Great Plains

This paper is based on a United States National Science Foundation funded project whose goal is to share resources for managing drought risks and encourage community participation in volunteer drought monitoring. Along the Great Plains of the United States, rural communities face "wicked" problems, such as population loss and economic decline [12,13]. The concentration of rural counties in the Great Plains complements the nation's densest concentration of agricultural and other resource-based industries [14]. These problems are historically interconnected to experiences with drought and the parallel impacts on farming and ranching operations [15]. For example, research in Oklahoma and New Mexico confirms that prolonged intense drought resulted in the large-scale selling of cattle, changes in crops planted, and decreased land cultivation [16]. Drought causes widespread economic losses and land degradation, and the teleconnections between rural

agricultural sectors and urban consumption can amplify those impacts (e.g., producing high beef prices nationally) [17,18]. Scientists predict that megadroughts, defined as unprecedented decadal and multidecadal drought conditions, will increase in the Southwest and Central Plains of North America in the coming decades [19]. Rural communities dependent on natural resources will need to enhance their capacity to adapt to the impacts of future droughts, which lie outside their coping range. The changing environmental landscape, demographics, and economies of the Great Plains demand a change in the 'why' and 'how' of investing in STEM education in rural America. To date, federal agricultural agencies have absorbed the vast majority of potential financial resources for rural places, making it difficult for communities to develop new areas of competitive advantage.

Additionally, in the face of a shifting rural-to-urban population, land grant universities struggle financially to provide the educational services that could benefit rural communities in sparsely populated and high-poverty areas [20]. This is particularly true for weather and climate information [21,22]. Rural places can be laboratories for social innovation, but this will require investments in rural communities' social and human capital [23,24].

Long-term experiences with drought have significant and long-lasting impacts on rural and small communities across the nation. Environmental damage (e.g., wildfires, crop loss, declining water supply, or invasive and nuisance species) cause stress to families and contribute to rural economic decline. Therefore, from a sustainability perspective, it is essential to engage with all stakeholders on ways to address drought. One such venue for engagement is at the intersection of emerging learning technologies and citizen science.

### 2.3. Citizen Science and Learning Technologies for Sustainability

Citizen science projects are important testing grounds for discoveries. Learning technologies can play a significant role (e.g., gamification to advance hydrology projects [25]) in bringing about or learning about novel solutions. This is affirmed by organizations such as UNESCO, when, in 2021, through its Science Commission, adopted an open science recommendation for global policy and regulatory agenda, highlighting the emancipatory potential of citizen science to collect, aggregate, and share data in ways that advance social and environmental resilience [26,27]. The technology for citizen science water monitoring exists and serves to ground truth satellite data and assists in emergency water management and transdisciplinary research about water challenges and extremes. In terms of drought, local monitors can help meteorologists predict extreme weather (e.g., flash droughts) and local farmers and ranchers to make sustainable production decisions. Still, the inherently technology-driven and data-driven relationship between citizen science and open science raises many unresolved questions about accessibility, transparency, and transdisciplinarity [28]. For example, the prevalence of mobile devices, remote sensing, wireless sensor networks, and other technologies have generated tremendous amounts of volunteered geographical data while raising important privacy, security, and ethics issues [29,30].

New technologies alone will not foster sustainability, but successful adaptations into educational practices can lead to local level actions. It is essential to look to the social and learning sciences to understand what motivates conservation behaviors. In the case of citizen science, areas such as hydrology and rural agricultural landscapes, for example, immersive technologies, can allow us to learn about the role of stress and risks of water insecurity as motivational drivers and influence the types of involvement from volunteers. In terms of drought, adaptive technologies can be helpful for co-produced data visualizations and planning tools [31–33].

The concatenation of learning technologies and citizen science moves the conversation beyond things, such as human sensors and crowdsourcing, to an understanding of the role of human behavior, cognition, and psychology in technology design. This paper focuses on using emerging learning technologies in informal settings that engage youth and family audiences in a virtual citizen science fair in Mozilla Hubs. We argue that purposeful instructional design and emerging technologies can advance ESD in many promising ways. Such ways include the creation of citizen science apps, the development of guided

field trips, gamification, role-playing, virtual fairs, data visualizations, or demonstrating processes that are otherwise hard to observe.

### 2.4. Immersive Technologies as Learning Tools

Interest in the use of virtual reality (VR), augmented reality (AR), and mixed reality (MR) is gaining momentum in education, businesses, and other industries. These emerging technologies, referred to as extended reality (XR) by [34], form what is commonly known as immersive technologies. Questions on how these emerging technologies can promote and impact learning have led to various new studies. Research has shown that XR provides a fun learning environment for learners and positively impacts their learning and behavior. In [35], the authors describe the experience of using XR in a full, rich, and immersive learning environment.

There are many educational benefits to integrating VR technologies with traditional teaching methods. For example, using VR in a blended learning environment allows for teacher-student discussion and is seen as an effective for of inquiry-based learning [36]. VR enables critical learning opportunities in an entertaining and engaging process [37]. A study by [38] discovered that an immersive and accessible VR environment allows learners to collaborate and socialize while learning. In addition, the study by [39] on a five-week VR course found that VR supports interaction and social presence and can include a broader range of communities. The study further showed that, during the length of the course, although there were technical difficulties, Mozilla hubs had no negative impacts on students' appreciation of the course. Interestingly, more students scored higher grades when VR was implemented. However, there was no explanation for the better grades achieved by students because the study was designed to measure the impact of VR usage.

Mozilla Hubs is an online VR platform that has been shown to increase engagement, but it has challenges in terms of use by older users. A study by [40] on team formation and communication using Mozilla Hubs found that enclosed and highly detailed spaces are preferred over large spaces. However, an open space with few details encourages interaction and collaboration. Younger participants learned the Mozilla Hubs controls faster and were more excited about using the platform than older participants [41]. A study by [39] discovered that even though there were challenges in using VR technologies such as Mozilla Hubs, challenges were not unique to the environment and were also faced by learners who were not experienced in using VR.

Additionally, the study found increased group collaboration, interaction, and engagement. The excitement VR has brought to many organizations and educational institutions about its use as a learning tool has allowed educators to be creative in how they use it to teach. Outside of school, however, VR can also be used as an informal science learning environment to teach drought and environmental monitoring.

### 2.5. VR as an Informal Science Learning Environment

Organizations and interest groups are finding ways to integrate VR in informal science learning (ISL) environments. In their study, [8] demonstrated that VR provides exciting opportunities in science education and public outreach (EPO) practices. In addition, they discussed that public engagement is less studied in EPO because engagement is broad and multidimensional. The study found that VR can facilitate tasks and lead to richer, more effective, engaging collaborations. A study by [42] focused on using VR as a tool for educators to address environmental problems, specifically the issue of raising public awareness about ocean acidification (OA) to the level of climate change awareness. Using VR can enhance learning experiences and understanding of unknown issues, such as OA, thereby addressing the challenge of making abstract concepts more concrete. Similarly, a study by [43] about creating immersive virtualization and experiences for abstract scientific concepts discovered that VR could simulate the effects of climate change on forests, weather patterns, tree species, distribution, and abundance.

*2.6. Designing Curriculum Using Mozilla Hubs*

Educational institutions are experimenting with integrating VR into their curriculum. For example, [44] integrated the Mozilla Hubs VR platform into a movie production course at the University of Gothenburg in Sweden. In a different study on usability, [41] discovered that learners found Mozilla Hubs easy to use and were satisfied with their experience. In addition, a study by [39,40] showed that there were increased interactions, collaboration, and engagement seen with learners using VR. Mozilla Hubs, in this case, allow for new ways of integrating emerging technologies with traditional teaching methods [37]. With proper guidance in designing and developing a virtual learning space, teachers and educators can create virtual learning environments that are engaging and entertaining for learners [38]. Eriksson [44] found that VR applications, such as Mozilla Hubs, have technical issues that make them more suitable for smaller groups than larger groups [38]. The study also found that uploading content in a virtual space can be challenging. Even though there are issues and challenges in creating and using VR learning spaces, studies show increased interest by educational institutions and interest groups to explore VR technologies to facilitate communication, group collaboration, and team-building [40]. By adopting emerging design practices of avatar-based VR social platforms, such as Mozilla Hubs, learners will be able to see significant embodied experiences, communication, and interaction in the virtual learning space [45].

Our literature reviews on AR, VR, and MR in formal and informal education demonstrate how VR can be used in learning and its impact on engaging stakeholders in community science. Results also reveal a need for further studies that focus on designing virtual learning spaces that are friendly to adult learners who expressed difficulties with being immersed in VR learning spaces. Our study draws on [37]'s research about VR integration with traditional teaching methods and 38]'s study on providing support and guidance on developing and designing virtual learning spaces that are fun and engaging. By creating a fun, engaging, and entertaining virtual learning space, educators can engage citizens in their process of learning about various citizen science projects using VR [37].

## 3. Theoretical Framework and Background

Underpinning the efforts of this project was a desire to use emerging technologies to communicate and educate rural communities about volunteer drought monitoring. Using a diffusion of innovation (DoI) framework, our research examines how librarians use the emerging technologies and adopt them into their library programs and spaces. DoI consists of five stages, which form the innovation–decision process. An individual (or decision-making unit or a cultural system) evaluates and decides whether to incorporate innovations into ongoing practices. The diffusion of innovation process illustrated in Figure 1 begins with the *knowledge stage* when an individual or a group becomes aware of the innovation and begins to understand how it functions. At the *persuasion stage*, a favorable or unfavorable opinion is formed based on the perceived characteristics of the innovation. At the *decision stage*, an individual (or decision-making unit) engages in activities that lead to a choice of whether to adopt or reject the innovation. In other words, adoption means that an individual or a decision-making unit does something different from what was being done previously. During the *implementation stage*, the innovation is put to use. Lastly, in the *confirmation stage*, the individual or decision-making unit seeks confirmation of the chosen decision to adopt or reject the innovation.

*Background*

This paper draws on research from the NSF AISL Innovations in Development project entitled "Spotty Rain Campaign: Enhancing the Capacity for Rural Libraries to Engage the Public in Drought Monitoring," which focuses on the design, development, and evaluation of informal science education (ISE) programs and educational media for use in rural libraries in drought-prone areas of the Great Plains. When it comes to citizen science weather monitoring, programs that connect to community problems (e.g., drought) raise

public interest and engagement and, in turn, scientific/climate literacy [46]. However, little is known about the capacity of rural libraries to promote and engage their local communities in citizen science programs [47]. The audiences for this project include public librarians and library staff (professionals) in rural and small communities of Oklahoma and the general public (adults and children) they serve. The project goals are to leverage rural librarians' professional skills and community knowledge to support local drought monitoring networks.

**Figure 1.** Five stages in the diffusion process.

Our project aims to encourage librarians to introduce citizen science processes and practices within the community dialogue and deliberation about drought. The Spotty Rain Campaign is primarily composed of the following strategies: (a) professional development webinars for rural librarians that introduce the resources necessary for training volunteer drought monitors and supporting citizen science practices and processes; (b) educational media (e.g., drought infographic booklet and poster series), and use of emerging technologies to improve public awareness, interest, and understanding about the role of science to inform drought adaptation; and (c) co-designed library-sponsored programs for public audiences.

Our project was initially envisioned to take part in a world where people can gather in a physical space. However, in March 2020, the COVID-19 pandemic forced public schools (K-12), universities, businesses, and organizations in the US and worldwide to shut down. Educators and professionals were forced to think of creative ways to stay engaged and work in isolation. At the same time, video conferencing tools such as Zoom and Microsoft Teams saw an increase in usage during the pandemic [38,44]. However, these communication tools have limitations. These limitations include the feeling of not being present, limited human interactions, disengagement, and fatigue, often caused by over-usage. This has led to an increased need for educators and professionals to find creative and innovative ways to engage with students, coworkers, stakeholders, and members of their communities [44].

This paper reports on the process of persuading both librarians and citizen science programs to adopt Mozilla Hubs as a virtual meeting space for library patrons. Among the stages of the DoI model, the persuasion stage is regarded as the most important. Up to 87% of adoption of an innovation is influenced by perceptions that potential adopters form toward an innovation [48]. Perceptions are formed based on five attributes listed in Table 2.

During the COVID-19 pandemic, the idea of virtual spaces for engaging the public gained a relative advantage. Librarians were seeking online learning that was compatible with the programs traditionally hosted and capable of serving the needs of youth and families now homeschooling. Similarly, citizen science programs searched for ways to share their digital educational resources and raise awareness about volunteer opportunities. Early in the project, demonstrations in Mozilla Hubs allowed librarians and citizen science program leaders to understand and navigate the Virtual Citizen Science Expo (VCSE) hall. This helped reduce the perceived complexity of virtual reality, and the trials were vital to creating buy-in and subsequent co-design efforts to build rooms for each partnering

organization. This paper describes how we deployed the Mozilla Hubs platform to host events for Citizen Science Month in April 2021. There was cross-promotion with our partner libraries and citizen science programs to increase the VCSE Hall's observability to youth, families, and educators.

**Table 2.** Perceived attributes of an innovation.

| Attribute | Explanation |
|---|---|
| Relative Advantage | Refers to the extent that an innovation is perceived as better than what an individual has or had before. If an innovation is seen to offer a greater relative advantage, then the rate of adoption increases. |
| Compatibility | Addresses whether or not an innovation is consistent with existing value structures and past experiences of a potential adopter. If an innovation does not conflict with existing value structures, then the possibility of adoption increases. |
| Complexity | Concerns perceptions that an innovation is difficult to understand and use. If an innovation is easy to use, then the possibility of adoption increases. |
| Trialability | Focuses on whether potential adopters can try out an innovation before fully committing to it. If an innovation can be tried out or experimented with on a limited basis, there is a greater chance of adoption. |
| Observability | Refers to the degree that the results of an innovation are visible to others. If results are observable and potential adopters can ask questions about it from friends or neighbors, adoption is easier. |

Source: Rogers, 2003.

## 4. Research Questions

The following research questions guide the current study:

RQ1: how does VR support learning about drought and environmental monitoring?

RQ2: what are the important design considerations for developing Mozilla Hubs rooms?

## 5. Method

This research employed a case study research method. In [49], the authors argue that, as an approach, a case study "facilitates exploration of a phenomenon within its context using a variety of data sources" (p. 544), and [50] adds that it can be characterized by a lack of control over unfolding events. As state earlier, the focus of this study was to explore the use of virtual reality as a learning space. Since the researchers had little control over the behaviors of the visitors in the space, and the data in which we are reporting were bound to a specific period, a case study approach is an appropriate method for which to present the data.

In this project, a virtual space was developed using Mozilla Hubs to celebrate Citizen Science Month in April 2021. The research team constructed a virtual space called the Virtual Citizen Science Expo (VCSE) in Mozilla Hubs, where visitors can join by using their mobile phones, computers, or virtual goggles. The VCSE was designed in partnership with the Southeast Oklahoma Library System (SEOLS), serving 15 small rural libraries in 7 counties in Southeast Oklahoma. We aimed to co-design and implement online programs that introduced citizen science processes and practices. Our goal was to produce "plug and play" programs focusing on weather and water projects for broader dissemination to other rural libraries.

The April VCSE had over 103 registered participants, including several educators with large student groups. This free, publicly accessible program utilized emerging AR/VR technologies to engage youth and families while meeting the virtual programming needs of librarians and citizen science programs in the era of COVID-19. The planning of Citizen Science Month events and codesign of the Mozilla Hubs rooms involved a team of 57 individuals overall. The team included rural librarians, state and national citizen science programs agencies, an advisory board, undergraduate and graduate students, and researchers across the domains of natural/water resources and emerging technologies.

These series of interactions with stakeholders such as librarians, youths, and citizen science vendors were captured using Snagit, 2022 version (a screen capture software). Our online observations were recorded using design notes, and finally, several screenshots of sessions in the VCSE rooms were captured. For this paper, we reviewed the chronological and technical evolution of the design. The goal was to derive practical and theoretical design considerations for using the Mozilla Hubs platform as a virtual learning space.

## 6. Results and Findings

This section illustrates how we used learning technologies for sustainable development. To share our findings, we will provide the reasons behind selecting the technologies used and share an illustrative case for how we used virtual reality to support our efforts to learn and teach about sustainability. Our efforts are to provide an illustrative case of how to advance the incorporation of VR as a learning tool and explain how VR could be incorporated into promoting and engaging learners in citizen science projects.

### 6.1. The Virtual Citizen Science Expo (VCSE)

The VCSE (https://hubs.mozilla.com/c4BXbbV/main-hall, accessed on 6 April 2022) was launched in April 2021 and consisted of different rooms that mimicked physical expo halls in the real world. The five areas comprising the VCSE included: (1) the main hall with instructions for using and moving around the Mozilla Hubs platform, (2) a lobby with directional signage, (3) Be a Drought Monitor rooms for each of our three citizen science project partners: the Community Collaborative for Rain, Hail, and Snow (CoCoRaHS), the National Drought Mitigation Center (NDMC), and the Oklahoma Blue Thumb, (4) the Southeast Oklahoma Library System (SEOLS) Backyard Explorer rooms (described in more detail below), and (5) a top floor with posters of different Oklahoma-based citizen science projects, for example, invasive species and soil sampling projects. During April, these citizen science program leaders participated in "Meet a Scientists in a Virtual World" events where they interacted in avatar and video chat form with visitors.

When learners join the Expo, they can select their avatar of choice and enter the virtual space. Avatars are virtual humans that have an essential role in virtual learning spaces as they support social life [45] and encourage communication and interaction in the virtual space [40]. In the VCSE, young scientists entering the virtual learning space as avatars were able to engage in conversations with others and real scientists. These interactions involve clicking and viewing 2D and 3D objects, such as posters, videos, websites, and models of insects and leaves that are integrated into the rooms in the VCSE.

### 6.2. Timeline and Purposeful Instructional Design Choices

As indicated in earlier sections, the Mozilla Hubs VR platform was the learning tool of choice used for designing the VCSE. This emerging technology was chosen because of its availability, affordability, and portability. Mozilla Hubs was compared to other VR platforms, such as Oculus and HTC Vive. Unlike the VR platforms that require a VR goggle to view scenes, Mozilla Hubs is a web-based VR platform that is free and accessible by anyone with a device that has a web browser and high-speed internet connection. This section details the different steps undertaken to design the space.

#### 6.2.1. Virtual Citizen Science Hall (VCSE)

The designing process of the VCSE main hall consisted of five steps. The first step of the design process was to select a scene. A scene in Mozilla Hubs is an existing design or template of a room that can be used to design a space for virtual social interaction, communication, and learning. In this case, a scene was selected and named main hall, as shown in Figure 2. The main hall scene was chosen because of its overall design, its modern style of architecture, and its openness that allows learners to explore the space freely.

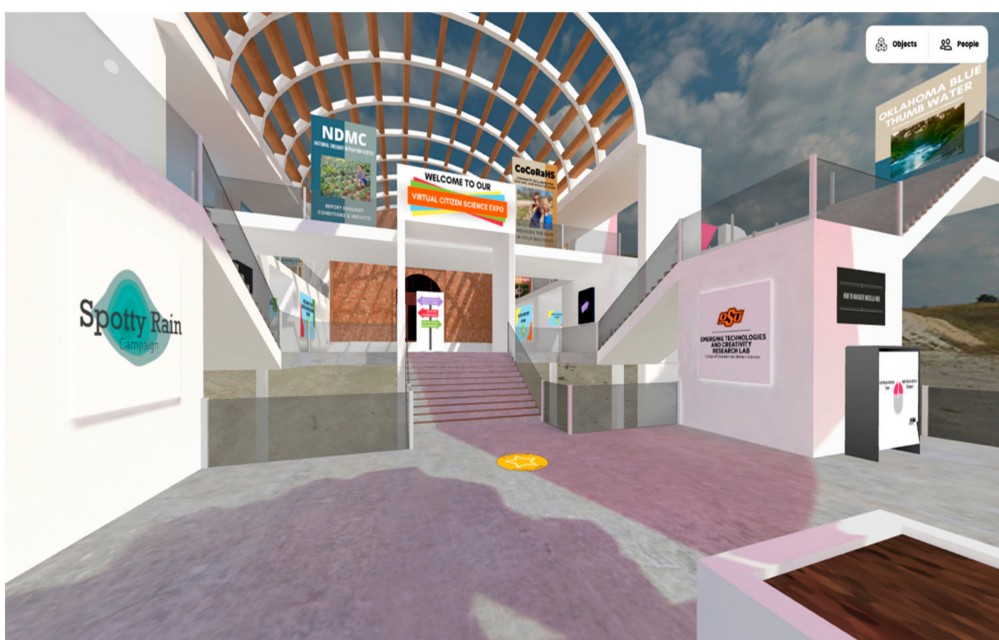

**Figure 2.** The main hall.

In the second step, banners, and logos instructions were added to the space. This included a welcome banner, the project and lab logos, and navigational instructions added at the landing spot. The landing spots in Mozilla Hubs are the areas where the user appears as an avatar when they join a room. In the third step, direction arrows and signs with names of projects were placed accordingly in the lobby and stairways to provide learners with information about the different citizen science projects. The signs also showed locations in the space while giving them a sense of direction as they navigated the virtual learning space. The fourth step included integrating room entrances in the main hall as shown in Figure 3. The entrances are screenshots of the actual virtual project room entrances, placed against the walls in the chosen sections of the main hall and linked to the corresponding citizen virtual rooms that are separate rooms from the main hall. For example, the NDMC entrance screenshot is linked to a Mozilla Hubs room called the NDMC room with a different website address. By clicking the NDMC room entrance, learners will be teleported out of the main hall and asked to join the NDMC room. In the final step, posters of various Oklahoma based citizen science projects were added to the top floor of main hall as shown in Figure 4 and linked to each of the organization's websites. Learners exploring the space can click the posters that will open a browser with the organization's website and provide the learners with information about the citizen science project and what the organization does.

6.2.2. Virtual Citizen Science Hall (VCSE)

In the designing of the CoCoRaHS (Figure 5), NDMC (Figure 6), and Blue Thumb (Figure 7) rooms, a "conference room" scene was selected for each of these organizations and branded with the organization's logo, banners, and choice of colors. The contents in each room consisted of information on what each organization does, information about local coordinators and how to volunteer, and social media contacts. Each of the branded rooms aimed to provide educational videos and various resources, such as how to measure drought, precipitation (rain, hail, and snow), and water qualities. Additional resources included interactive games and cartoon videos that teach learners about drought monitoring and decision-making.

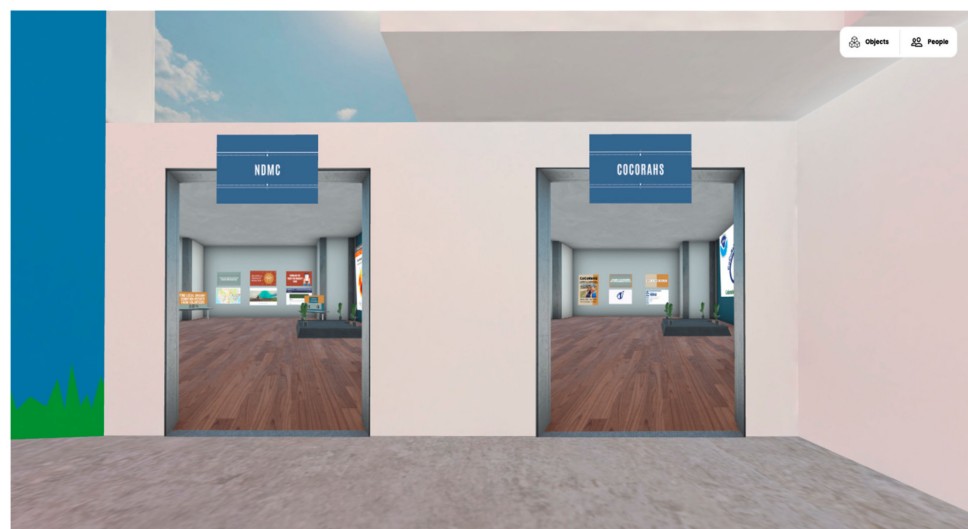

**Figure 3.** Entrances to NDMC and CoCoRaHS rooms.

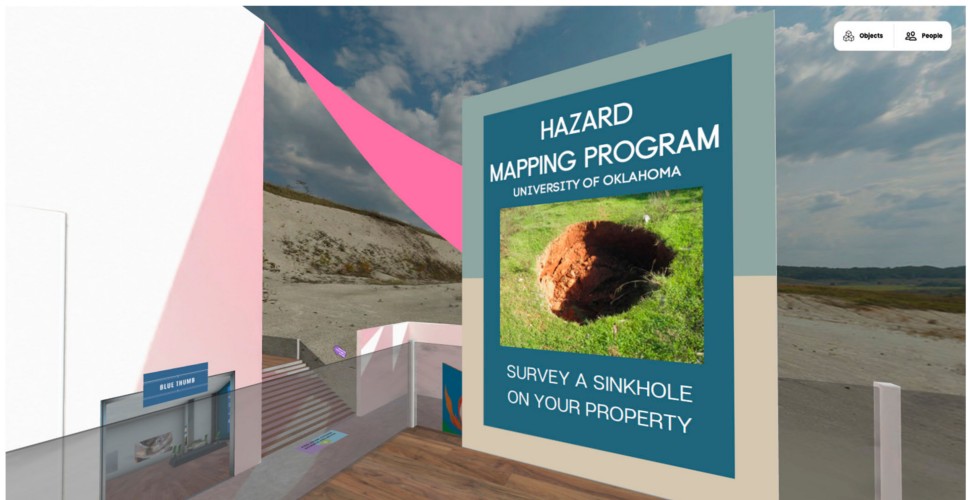

**Figure 4.** A poster of the hazard mapping program on top of the main hall.

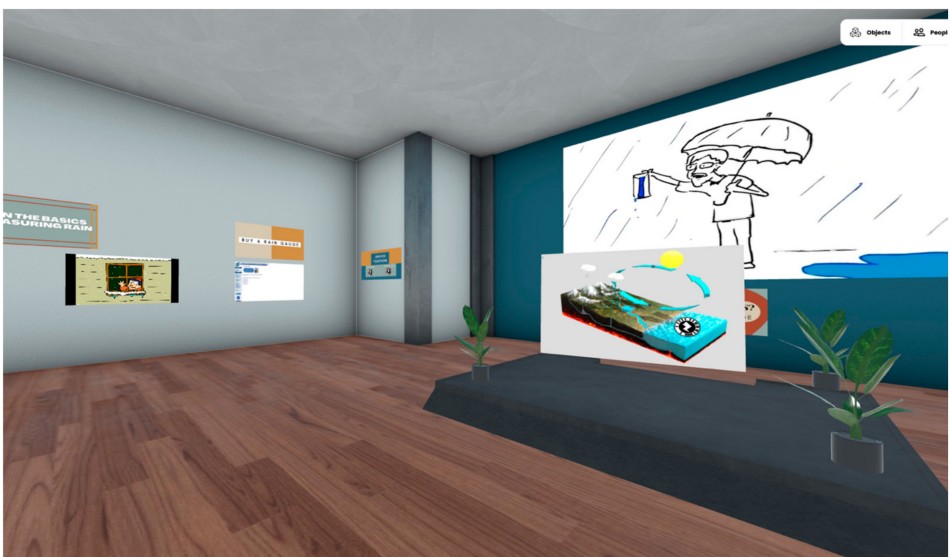

**Figure 5.** Inside the CoCoRaHS room.

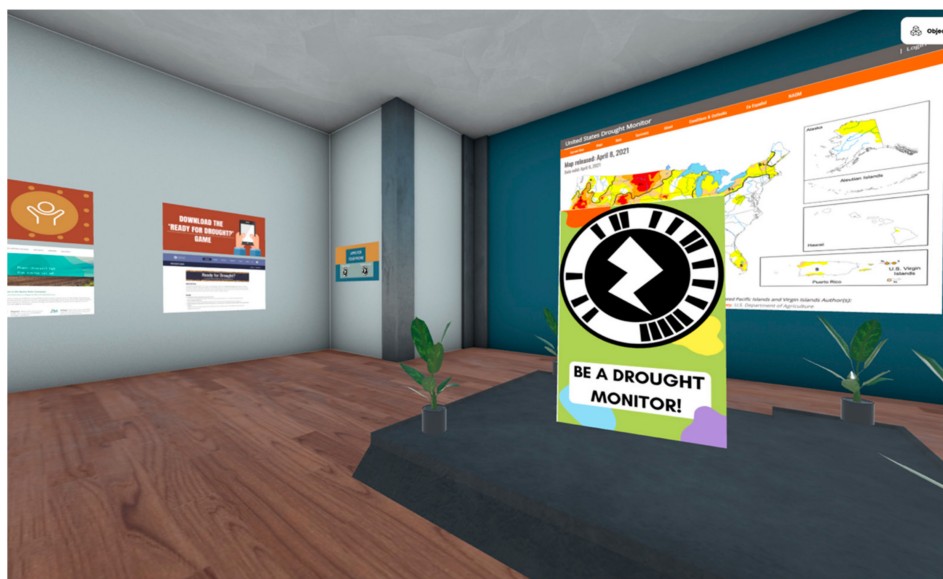

**Figure 6.** Inside the national drought mitigation center (NDMC) room.

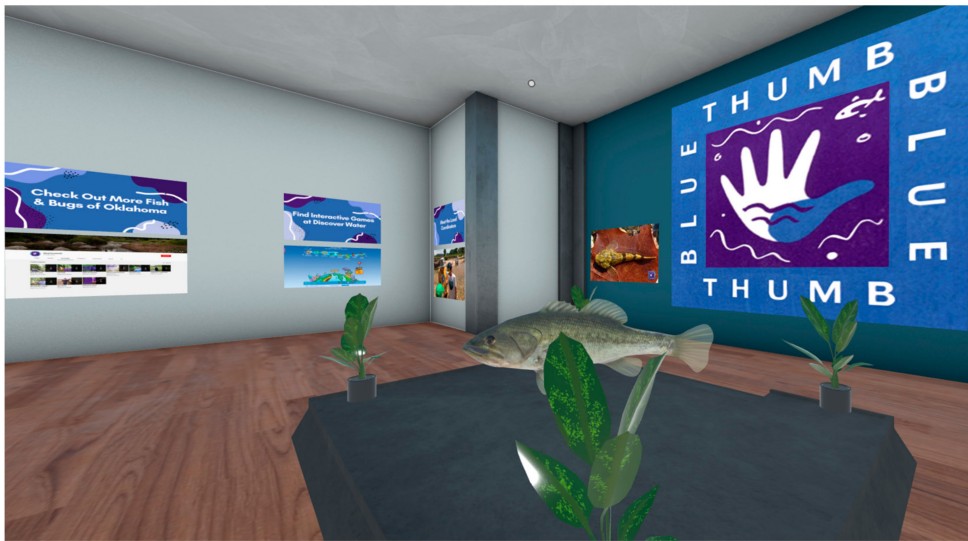

**Figure 7.** Inside the blue thumb room.

The objective of this project is that, as learners participate and interact in these citizen science rooms and as they learn about the different citizen science projects, they will develop interests in these projects and get involved by volunteering in some of the activities. In order to volunteer in these organizations, learners will sign up through the information provided in the space or contact the local coordinators of these organizations. Finally, learners can stay up-to-date with the latest volunteer information, projects, and activities by following the organization through their social media.

6.2.3. Library Program Rooms

Several rooms in the VCSE were designed to support the SEOLS Backyard Explorer (BYE) program. The BYE curriculum includes a video and a logbook challenge introducing youth and families to nature inquiry and observation. We created zapcodes to put on each logbook that included AR 3D models of leaves, insects, and the weather cycle on users' phones. Zapcodes are codes circular in shape with a lightning bolt in the middle. In order to use these zapcodes, the Zappar app needs to be installed on a user's mobile device. In addition, Mozilla Hubs rooms were created related to the phenomena described in the

two BYE videos, leaves and insects. Like the CoCoRaHS, the NDMC, and the blue thumb rooms, each BYE room was designed with the room theme. In the front section of each room (Figures 8 and 9), an introduction video by a SEOLS librarian. The video informed learners what the room is about and the types of information and resources to find in the space.

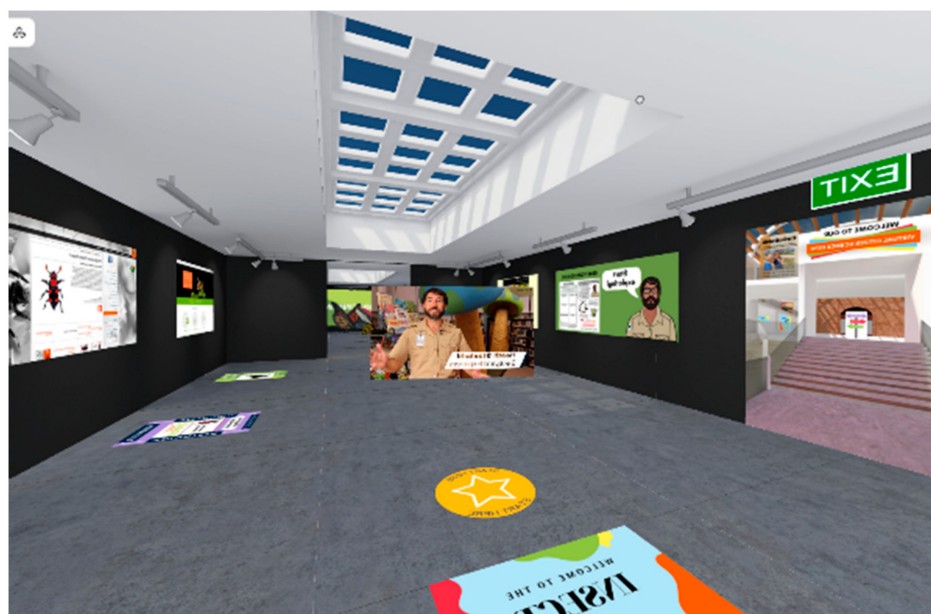

**Figure 8.** Leaf Room–Front Section.

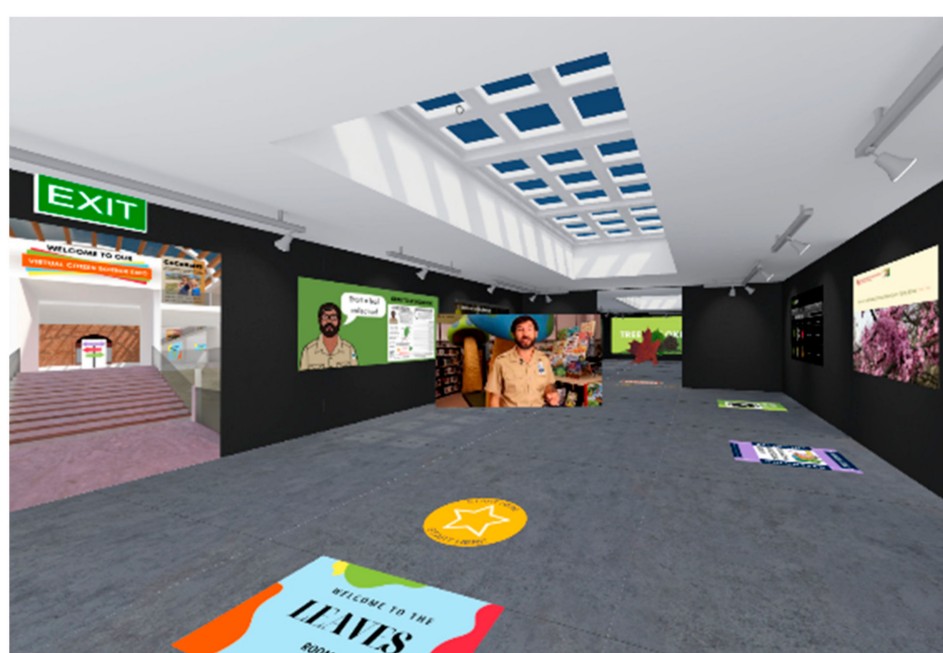

**Figure 9.** Insect Room–Front Section.

Additionally, learners will find educational resources such as the logbook challenge, ID key, ID guide, and Cool Tools. The spaces also have local park information where users can go to complete their logbooks challenge and information about how to volunteer in citizen science projects and environmental monitoring activities. In the back section of the rooms (Figures 10 and 11), learners will find pictures and 3D models of different species of leaves and insects described in the BYE videos. These themed rooms provided

educational resources to backyard explorers and engaged them with the SEOLS library BYE program. As learners participate in the library's BYE program and develop an interest in how to explore the environment, they are encouraged to volunteer for related citizen science projects and visit their local library to find resources for these projects. Table 3 provides additional details on the components of the hub space as well design considerations.

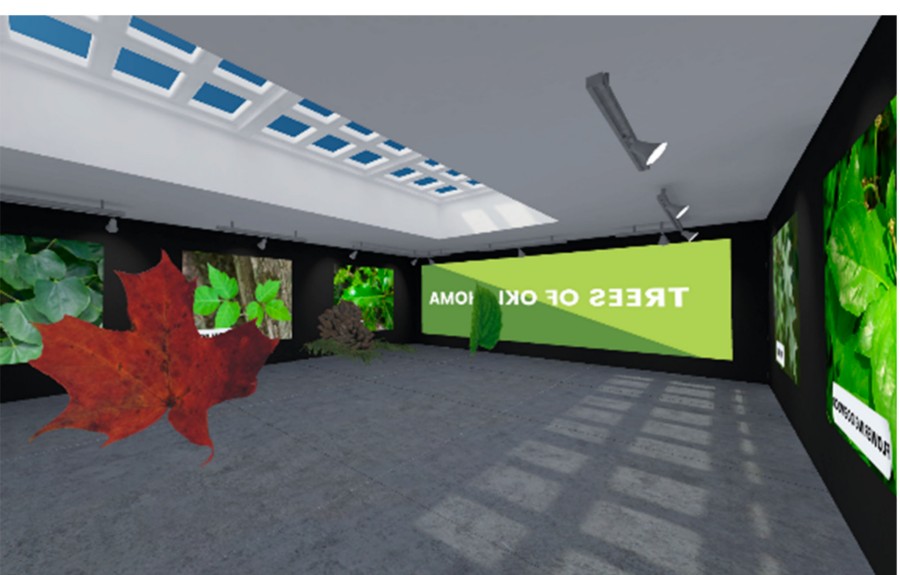

**Figure 10.** Leaf Room–Back Section.

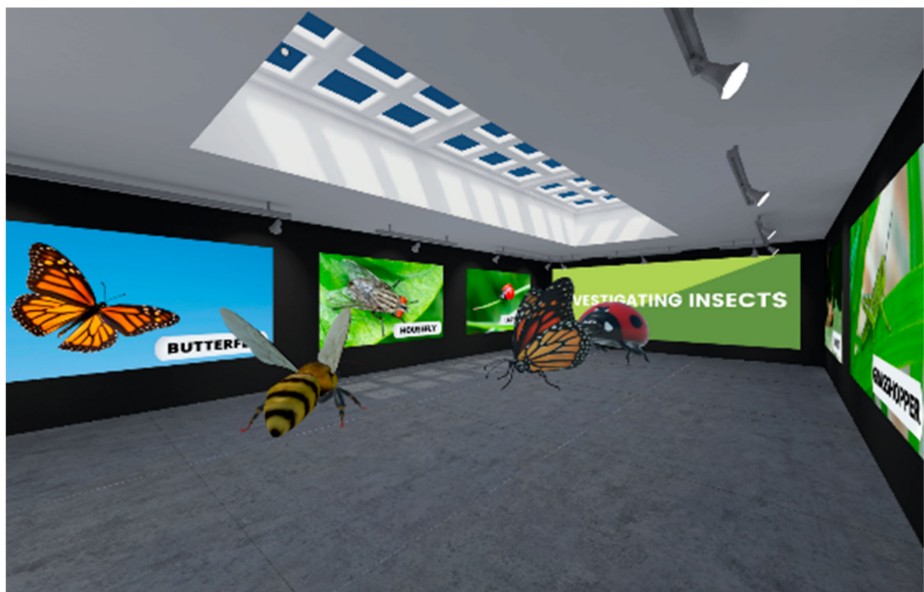

**Figure 11.** Insect Room–Back Section.

**Table 3.** Matrix of VCSE Components by Design Factors.

| VCSE Components | Summary of Learning Activity | Environmental Education Goals | Instructional Design Choices | Technical Considerations |
|---|---|---|---|---|
| Backyard Explorer Rooms | Access the BYE logbook challenges. View 3D models and images of species from BYE videos. | Explore and deploy educational resources for nature inquiry and identification. Discover a related citizen science project. | Adding welcome video 3D models and images of species discussed. | Video file format: MP4. 3D Models size: 10 MB or less. Image size: 1240 × 1240 px or less. Use of high speed internet. Use individual devices instead of multiple users to a station or device. |

**Table 3.** *Cont.*

| VCSE Components | Summary of Learning Activity | Environmental Education Goals | Instructional Design Choices | Technical Considerations |
|---|---|---|---|---|
| Meet the Scientist Events | Ask real scientists about weather and water. | Learn about drought and the importance of environmental monitoring. | Open discussions. Question and answer session in the rooms. | Using the camera feature to show your face instead of an avatar head. |
| Be a Drought Monitor Rooms | Explore educational resources, including volunteer maps. Watch videos to learn about drought, water cycle, and climate. | Discover volunteer opportunities. Learn how to measure and report precipitation, as well as drought condition reports. | Educational videos and resources related to water and drought. Volunteer maps. Contact and social media profiles. | Video file format: MP4. 3D Models size: 10 MB or less. Image size: 1240 × 1240 px or less. Use of high speed internet. Use individual devices instead of multiple users to a station or device. |
| AR Logbook/ BYE Challenges | Participate in BYE-themed challenge (leaf collection, insect scavenger hunt, or weather watching) Scan logbook for 3D model. | Learn about the processes and practices of nature inquiry and observation | Downloadable PDF logbook for printing and use. Use of zapcodes. Interactive AR models. | Easily accessible from different devices. |

### 6.2.4. Technical Lessons Learned about Design in Mozilla Hubs

Designing a VR space is an iterative process that is driven by users' feedback about their experiences in the space and expert guidance. This iterative and cyclical process is consistent with DBR characteristics outlined by [51]. The feedback can be in the form of direct communication by users about what they experienced in the space or through field notes and video recordings taken by researchers. The feedback provided can be used to redesign the virtual spaces.

There is interest in VR and how it can support learning, teach abstract concepts, engage the public in citizen science and environmental monitoring, promote public participation and encourage volunteering. Based on the feedback received during the design process of VCSE, we were able to learn valuable lessons about how VR can be used as a learning tool for teaching and engaging learners in different fields.

There were various lessons learned during the design of the VCSE and after the learning space was used. During the design process, the lessons learned were, for the most part, based on trial and error and the frustration faced by the design team. This is in large part because there are limited design guide documents for those who wish to engage in creating a Mozilla Hubs space. Perhaps the first and most apparent discovery is that before undertaking the design process in the hub, one must have some familiarity and working knowledge of designing graphics, multimedia, and 3D spaces. Such an understanding must include an appreciation that designing virtual spaces is a reiterative process that takes time. Once inside the Mozilla hubs, one quickly learns that it is necessary to know how to use Mozilla Spoke, the scene editing tool for integrating educational materials into learning spaces. In other words, while Mozilla Hubs is the platform in which users interact, Mozilla Spoke can be viewed as a tool for designing the space. Another lesson that one gains from being in the platform is that the best 3D models are 10 MB or less, and the best images are 1024 × 1024 px in size.

Once the VCSE was made public, and users could come into the space, there were additional lessons learned based on our interactions and observations. We frame the lessons learned after VCSE based on the usability, utility, and aesthetic constructs articulated by [52] about designing collaborative learning environments. When it comes to usability, one "is concerned with whether a system allows for the accomplishment of a set of tasks in an efficient and effective way that satisfies the user" (p. 50). Utility refers to how the system functions, as in whether it actually works. Aesthetics is concerned with how the tool "may appeal to and benefit the users, in a way that it absorbs the user within the interaction itself" (p. 52).

The following lessons were learned when applying the aforementioned constructs to the Mozilla Hubs space. In terms of usability, users could navigate the space using their

personal computers and mobile devices. Initially, the users were disoriented for moments, but once the controls were learned, the users were able to move around the space showing a preference for using keyboards to navigate.

One major issue concerned the utility construct. Connectivity appears to be the main issue for the system's function. Some users had difficulty connecting to the VCSE, especially when using older mobile devices. There were also issues with hearing others speak in the space unless their avatar was near the speaker or when there were too many users using the same device or speaking simultaneously. Another issue voiced concerns the privacy and access of the spaces and whether it was safe for the young learners. Some audiences were uncomfortable knowing that the VCSE is an open space that anyone can access.

The construct that provided the most lessons was aesthetic. Users between the ages of 8 and 13 had fun engaging in this virtual space, while the audiences from 5 through 7 years old mostly watched because they were unsure what to do in the space. The users found the idea of using Mozilla Hubs as a learning space very exciting and saw themselves using it in the future. Adult users had questions about the possibility of using Mozilla Hubs to meet and teach online classes, teach abstract concepts, and if these virtual spaces can replace physical learning spaces. Overall, it appears that users found Mozilla Hubs very interesting and engaging. Users also reported that experiencing Mozilla hubs gave them new ideas on how to teach and use VR in the future.

## 7. Discussion

Restrictions and guidelines imposed globally by governments during the COVID-19 pandemic limited public engagement in citizen science and environmental monitoring activities. This has resulted in educational institutions and citizen science organizations exploring new and creative ways to engage their communities in activities such as environmental monitoring. One of these new and creative ways is to use emerging technologies such as virtual reality (VR), to engage the communities in citizen science and environmental activities, promote public participation, and increase volunteering. In this paper, Mozilla Hubs, a social VR platform developed by Mozilla, was used to design the Virtual Citizen Science Expo (VCSE). This virtual learning space was used for engaging communities in various citizen science projects and environmental monitoring activities while promoting participation and encouraging volunteering.

The research on public engagement through immersive learning spaces in citizen science is still emerging. The creation and implementation of the VCSE address the paucity in this area of study. VR as a learning tool can provide a pathway to engage communities in citizen science and environmental monitoring activities, encourage public participation, and allow learners to create a deeper understanding of environmental issues [33,42,43]. With environmental concerns on the rise, VR can be used to explore ways to understand issues that affect participants' attachments to places [33], create an understanding of concepts that are difficult to understand [42], and provide learners with the ability to project the future and the past understanding of the environmental issues [43]. In addition, VR environments can be used to create environmental experiences and promote citizen science. It can avail new learning opportunities about the environment [33]. It can be used to teach abstract concepts and to develop new knowledges [42]. Therefore, VR presents a new and creative way to engage the public in citizen science and environmental monitoring activities, promote public participation and volunteering efforts, and support the creation of past, present, and future environmental experiences [43]. VR holds promise for future use in promoting public engagement in citizen science and environmental monitoring.

The limitation of the VCSE included getting enough visitors to join the virtual learning space; the number of visitors cannot exceed twenty-five participants at a specific time without causing lag and latency issues in navigating the space.

## 8. Conclusions

This paper describes an iterative design process in collaborating with citizen science programs and a rural library system. The VCSE engages youth and families in learning about environmental monitoring and volunteer opportunities to participate in science. The results inform future uses of VR to advance environmental education and other efforts to employ Mozilla Hubs for informal science learning. Overall, we argue that we can advance ESD in many promising ways with purposeful instructional designs and emerging technologies. Beyond creating citizen science apps, we can develop immersive guided field trips, gamification, role-playing, virtual fairs, data visualizations, or new ways of demonstrating processes that are otherwise hard to observe. Our future research will further explore how engagement occurs in virtual learning spaces, especially during intergenerational conversations, how visual designs impact learning motivation, and ways to engage in the co-creation and co-design of spaces that promote learning about environmental issues.

**Author Contributions:** Conceptualization, T.I.A. and N.M.C.; methodology, T.I.A. and N.M.C.; software, C.A.; validation, T.I.A., N.M.C., A.I. and C.A.; formal analysis, T.I.A., N.M.C., A.I. and C.A.; investigation, T.I.A., N.M.C., A.I. and C.A.; writing—original draft preparation, T.I.A., N.M.C., A.I. and C.A.; writing—review and editing, T.I.A., N.M.C., A.I. and C.A.; visualization, C.A.; supervision, T.I.A. and N.M.C.; project administration, T.I.A. and N.M.C.; funding acquisition, T.I.A. and N.M.C. All authors have read and agreed to the published version of the manuscript.

**Funding:** This material is based upon work supported by the National Science Foundation under Grant No. 1811506.

**Institutional Review Board Statement:** The study was conducted in accordance with the Declaration of Helsinki, and approved by the Institutional Review Board of Oklahoma State University (protocol code ED-18-52, 25 April 2018) for studies involving humans.

**Informed Consent Statement:** Informed consent was obtained from all subjects involved in the study.

**Data Availability Statement:** Not applicable.

**Conflicts of Interest:** The authors declare no conflict of interest.

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
