# Peer review of "The Virtual Citizen Science Expo Hall: A Case Study of a Design-Based Project for Sustainability Education"

_sustainability, doi:10.3390/su14084671_

Round 1

Reviewer 1 Report

The abstract should contain information regarding the following points: problem background, materials & methods, attained results, and conclusions.

In the introduction section, the problem statement of the study is not clearly identified. 

The exact contributions should be highlighted in the introduction section.

An organization section that describes the upcoming sections of the article must be added at the end of the introduction section.

The literature review is not covering the recent research papers in this field.

It is necessary to add a block diagram that explains the implementation specifics of the proposed methodology.

Although the used methodology has advantages, the authors should state its shortcomings as well.

The future works are not presented clearly.

Author Response

Thank you for your time to review our article. Please find the attached document containing our responses.

Reviewer 2 Report

This paper has provided an interesting topic for environmental education, and application of learning technology. And the results of Virtual Reality also delivered the different views from perspective of information technology.

Below, some suggestions for this paper as:

Table 1 is not suitable in the Section 1 Introduction. Because, that is a definition for learning technology, it should be in the Section 2 Literature Review.

Since this paper want to explore an information technology for the environmental education, however, the Section 2 Literature Review has no any information talking about Environmental Education System. Or VR/AR how to apply into environmental education.

In the Section 3, Figure 1 has appeared twice. Line 232 and line 234.

What is meaning in line 276? 1Source: Rogers, 2003. That is not a format as reference, Author should double check. And if cite this reference, should provide page number.

Section 5 Research Question should be deleted, the questions should be changed to Section 1.

Section 6 Results, the Figures’ number has problem, no Figure 8. Authors should check all Figures’ number again, especially from Figure 8 to Figure 12.

What is meaning in line 464? The words “Mozilla Hub VR (VCSE)” should be deleted.

Section 5 Conclusion should be Section “8.”

The Conclusion is not well. Because, it does not provide limitations and future works.

Totally, this paper is more specific on an application topic for learning technology. It is less parts on environmental education, but more parts on the presentation of VR/AR. 

Author Response

(The authors gave the same response as above.)

Reviewer 3 Report

Dear Authors,

The study uses a design-based project for sustainability education at the Virtual Citizen Science Expo Hall. It’s pretty interesting, owing it could be published, suggest authors clarify below question,

So many VR platforms, for more contributions, suggest authors compare different VR platforms and address why the study chose the Mozilla Hubs platform.

 The research questions section should move before the method, the study’s framework would be better.

Thank you.

Author Response

(The authors gave the same response as above.)

Round 2

Reviewer 2 Report

This paper I read the second, the revised version has better than the first time I read.

Only one thing is the Figure 1, in my personal suggestion, please do not use color to show what you want to say, because this paper is academic article, not a magazine color in here is not necessary.

Author Response

Dear reviewer,

Thank you for taking the time to review our manuscript yet again.

Comment: "Only one thing is the Figure 1, in my personal suggestion, please do not use color to show what you want to say, because this paper is academic article, not a magazine color in here is not necessary."

Response: We have changed the graphic from a colour version so that it is now in black and white/grayscale.

Best regards,